# How to Choose a Shunt for Patients with Normal Pressure Hydrocephalus: A Short Guide to Selecting the Best Shunt Assembly

**DOI:** 10.3390/jcm10061210

**Published:** 2021-03-15

**Authors:** Juan Sahuquillo, Katiuska Rosas, Helena Calvo, Aloma Alcina, Dario Gándara, Diego López-Bermeo, Maria-Antonia Poca

**Affiliations:** 1Department of Neurosurgery, Vall d’Hebron Hospital Universitari, Vall d’Hebron Barcelona Hospital Campus, Passeig Vall d’Hebron 119-129, 08035 Barcelona, Spain; krosas@vhebron.net (K.R.); hcalvo_rubio@vhebron.net (H.C.); aalcina@vhebron.net (A.A.); dfgandar@vhebron.net (D.G.); diego.lopez@vhebron.net (D.L.-B.); pocama@neurotrauma.net (M.-A.P.); 2Neurotrauma and Neurosurgery Research Unit, Vall d’Hebron Institut de Recerca (VHIR), Vall d’Hebron Hospital Universitari, Vall d’Hebron Barcelona Hospital Campus, Passeig Vall d’Hebron 119-129, 08035 Barcelona, Spain; 3Department of Surgery, Universitat Autònoma de Barcelona, 08193 Bellaterra, Spain

**Keywords:** differential-pressure valves, hydrocephalus, normal-pressure hydrocephalus, fluid mechanics, shunt overdrainage, antisiphon device

## Abstract

Most patients with hydrocephalus are still managed with the implantation of a cerebrospinal fluid (CSF) shunt in which the CSF flow is regulated by a differential-pressure valve (DPV). Our aim in this review is to discuss some basic concepts in fluid mechanics that are frequently ignored but that should be understood by neurosurgeons to enable them to choose the most adequate shunt for each patient. We will present data, some of which is not provided by manufacturers, which may help neurosurgeons in selecting the most appropriate shunt. To do so, we focused on the management of patients with idiopathic “normal-pressure hydrocephalus” (iNPH), as one of the most challenging scenarios, in which the combination of optimal technology, patient characteristics, and knowledge of fluid mechanics can significantly modify the surgical results. For a better understanding of the available hardware and its evolution over time, we will have a second look at the design of the first DPV and the reasons why additional devices were incorporated to control for shunt overdrainage and its related complications. We try to persuade the reader that a clear understanding of the physical concepts of the CSF and shunt dynamics is key to understand the pathophysiology of iNPH and to improve its treatment.

## 1. Background

Hydrocephalus is the most frequent cause of acute and chronic intracranial hypertension both in children and adults and is the condition that pediatric neurosurgeons treat most frequently. Despite the popularity of endoscopic third ventriculostomy in the neurosurgical armamentarium, most patients with hydrocephalus are still managed with the implantation of a cerebrospinal fluid (CSF) shunt in which the CSF flow is regulated by a differential-pressure valve (DPV). The first generation DPVs were introduced in the 1950s and most of the valves designed soon after World War II are still used today, with minor modifications [1,2]. In the last two decades, shunt technology has evolved significantly and in addition to the conventional DPVs, today we have at our disposal “programmable” valves, flow-regulated valves, antisiphon and gravitational devices, and many shunt assemblies with significantly different hydrodynamic profiles. In a review conducted by Aschoff in 2017, about 160 different shunt designs were available on the market [1] distributed by twelve manufacturers worldwide [3].

Until recent years, the most complex problem neurosurgeons faced was to choose among a low-, medium- or high-pressure DPV [3]. The wide range of designs and functional complexity of shunt assemblies currently available allow reduction of the complications of shunt over- or under-drainage and to improve the long-term outcome of patients with hydrocephalus. However, the information provided by manufacturers regarding the different types of valves is scarce and somewhat misleading, and basic fluid mechanics is not part of the standard training of neurosurgeons. In acute posthemorrhagic or tumoral hydrocephalus, any simple DPV available on the market will work but, in complex clinical scenarios, such as in patients with pseudotumor cerebri, arachnoid cysts, negative-pressure hydrocephalus, slit-ventricle syndrome, hydrocephalus in the pediatric population, and in patients with idiopathic “normal-pressure hydrocephalus” (iNPH) syndrome, the appropriate selection of shunt hardware is crucial for obtaining the best possible outcome and reducing shunt-associated complications.

Current shunt technology allows for selection among a wide variety of hardware and adapts it to patients’ age, height, weight, clinical, and neuroradiological characteristics. This selection is especially relevant in patients with iNPH in which the shunt selection is essential to reduce both the rate of nonresponders and shunt-related complications. In pediatric neurosurgery, the goal of selecting the most adequate shunt hardware is to reduce the number of shunt-failures that infants and children will have until they become adults, an important factor in obtaining their best long-term outcomes and social integration.

Our aim in this review is to discuss some basic concepts in fluid mechanics that are frequently ignored but that should be understood by neurosurgeons to enable them to choose the most adequate shunt for each patient. We present data, some of which is not provided by manufacturers, which may help neurosurgeons in selecting the most appropriate shunt. To do so, we focus on the management of patients with iNPH, as one of the most challenging scenarios, in which the combination of optimal technology, patient characteristics, and knowledge of fluid mechanics can significantly modify the surgical results. Although our statement seems to be contradictory with the lack of evidence that any valve or shunt assembly is superior in managing hydrocephalus [4], we hope to persuade the reader of the need to reconsider the old concepts regarding DPV.

## 2. How Neurosurgeons Select a Shunt?

In brief, a modern CSF shunt has five major components: (1) a ventricular catheter, (2) a reservoir, (3) a valve system, (4) an antisiphon (ASD)/gravity compensating-device, and (4) a distal catheter [5,6]. Because the designs of the proximal and distal catheters are quite constant among manufacturers (with internal diameters (ID) that are usually greater than 0.80 mm) their relevance is of minor interest. However, the valve design and the incorporation of an ASD/gravity-compensating accessory, whether integrated in the valve or inserted in series, significantly modifies the hydrodynamic performance of a shunt. In an excellent but undercited paper about shunts, Turner stated that “neurosurgeons approach the choice of hardware for use in a ventriculoperitoneal shunt in much the same way they buy a car” [5]. In his own words, neurosurgeons use “…style, past experience, brand loyalty, advertising, comfort, training, and a little science to choose a combination of hardware and surgical procedure” [5]. This caustic remark reflects the reality of shunt surgery in many neurosurgical departments in which the hydrodynamic knowledge of residents and staff is suboptimal most of the time, and shunt surgery is considered a nonglamorous and technically simple surgical procedure that is frequently assigned to residents or junior staff. Frequently, the selection of hardware for implantation is based on the stock each neurosurgical department owns that is still generally based in the simple and misleading concept of low-, medium- and high-pressure DPVs.

For a better understanding of the available hardware and its evolution over time, we will have a second look at the design of the first DPV and the reasons why additional devices were incorporated to control for shunt overdrainage and its related complications [7]. We try to persuade the reader that a clear understanding of the physical concepts of the CSF and shunt dynamics is key to understand the pathophysiology of NPH and to improve its treatment.

## 3. Differential-Pressure Valves: The First Generation

One of the first attempts in managing hydrocephalus was reported by Dandy who proposed third ventriculostomy to treat obstructive hydrocephalus and the extirpation/coagulation of the choroid plexus in communicating hydrocephalus [8]. Early attempts to divert the CSF from the cerebral ventricles with saphenous grafts to the sagittal sinus or to the heart atrium with simple tubes made of plastic, rubber, or metal were unsuccessful because they become clogged by blood when the venous pressure was increased by any Valsalva maneuver [8].

The discovery of the silicone elastomers in 1943 was a significant milestone in medicine. Silicone is an inert material with low interaction with biological tissues that sustains high temperatures and therefore, it can be sterilized [9]. The first documented attempt to use an internal shunt with a unidirectional valve to avoid reflux was reported by the neurosurgeons Nulsen and Spitz in 1952 in cooperation with scientists from Dow Corning, a USA company specialized in silicon-derived polymers [9,10]. Later, Spitz implanted a unidirectional valve designed by Holter for his own son as documented in a captivating history by Baru et al. [11]. In the patent filed in 1956 by Holter and Nulsen, the reported purpose of the unidirectional valve was “the prevention of reverse flow and providing for a pumping operation for insuring that the device is functioning properly” [12]. The first valve model consisted of two in-series silicon tubes with lateral slits encased within a sleeve (Figure 1). The slits in the silicone acted as a DPV by allowing flow through them when differential pressure was ≥30 mmH_2_O (2.2 mmHg) [12]. The first DPV valves were introduced in the early 1950s by a reduced group of neurosurgeons in the United States [2] and the designs made by Pudenz and Holter were the first designs used for the treatment of hydrocephalus [2,8].

## 4. Basic Fluid Mechanics Required to Understand Shunts and Valves

In any nonelastic tube, liquid flow is driven by the pressure difference between the inlet and outlet. This was the remarkable finding that Poiseuille found in the experiments he designed to understand the laws governing blood flow in the microcirculation [13]. Poiseuille’s law describes the relationship between flow (Q) and pressure of any Newtonian liquid circulating through rigid tubes with nonturbulent (laminar) flow [14]. This equation is frequently used in physiology, engineering, and biology:(1)$$Q=\frac{\left(P_{1}-P_{2}\right)\pi R4}{8\eta \iota },$$

In Equation (1), P_1_ and P_2_ are the pressures at the inlet and the outlet of the tube, respectively, R is the radius of the tube, η the viscosity of the fluid, L the length of the tube, and π and 8 are constants. The viscosity of CSF is close to the viscosity of water, which is 0.7–1 mPa-s at a temperature of 37 °C [15]. Q in a rigid tube is directly proportional to P_1_–P_2_ and inversely proportional to the resistance of the tube that depends on R, η, and L. However, in any shunt assembly, the distal tube is only a part of a more complex design and therefore the total resistance of the shunt (R_shunt_) is always higher than the resistance of the distal catheter.

To address this issue, the analogy between steady-state electrical current flow in an electric circuit and the flow of an incompressible fluid in a nonelastic tube is helpful. In an electric circuit, the electric flow is defined by Ohm’s law that when applied to fluid dynamics can be restated:(2)$$Q=\frac{\left(P_{1}-P_{2}\right)}{\mathrm{Rshunt}},$$
in which (P_1_–P_2_) is the inlet–outlet differential pressure and R_shunt_ the total resistance of the shunt assembly. In an electrical circuit in which resistors are connected in series, the total resistance (impedance) will be calculated by adding up the impedance of the individual resistors: R = R_1_ + R_2_ + R_3_ +…. If an electric circuit has three in series resistors with an impedance of 8 Ω, 3 Ω, and 2 Ω respectively, the total impedance of the circuit will be 13 Ω. In a similar way, the total resistance of a shunt (R_shunt_) is calculated by adding the resistance of all the individual components in the shunt assembly:R_shunt_ = R_vent_ + R_dist_ + R_value_ + R_grav_,(3)

In Equation (3), R_shunt_ is obtained by adding the individual resistances of the ventricular catheter (R_vent_), the distal catheter (R_dist_), the resistance of the valve (R_valve_), and that of any additional device added in series to compensate for gravitational changes (R_grav_). The normal resistance to CSF outflow (R_out_) in healthy individuals ranges from 5 to 10 mmHg/mL/min, and this resistance increases with age [16,17]. When a patient with hydrocephalus undergoes a shunt implantation, both the shunt and the residual absorption capacity of the patient do work in parallel, and therefore, to reproduce the normal physiology the total resistance of both working together should be close to these physiological values.

## 5. Which Pressures Govern Cerebrospinal Fluid Flow in an Implanted Shunt Assembly?

In vitro, the flow across a shunt assembly, as defined in Equation (1), is directly proportional to the driving pressure, which is the difference between the inlet (P_1_) and outlet pressures (P_2_) and inversely proportional to the total resistance of the shunt (R_shunt_) as defined in Equations (2) and (3).

For a better understanding of the CSF dynamics across a shunt, it is useful to use the concept of “shunt perfusion pressure” (PP_shunt_) introduced by Rayport and Reiss [14], a term familiar to any medical student learning the regulation of blood flow through the capillary beds of any organ. Rayport and Reiss defined PP_shunt_ in VA shunts as the algebraic difference between the intraventricular pressure (IVP) and either the closing pressure (CP) of the DPV valve or the atrial pressure, “whichever is larger” [14]. In 1972, Fox et al. modified the PP_shunt_ concept by introducing the gravitational-induced hydrostatic pressure (HP) in the algebraic definition (Figure 2) [18]. HP is the gravitational-induced pressure generated by the column of CSF fluid in the shunt when the patient is standing or sitting and equals the difference in height between the tip of the ventricular catheter and the distal tip of the shunt. This HP is negligible when the patient is recumbent but can increase to 50 to 70 cmH_2_O (~37–50 mmHg) when a medium-height shunted patient is sitting or standing (Figure 3).

As shown in Figure 2, IVP and the HP within the shunt contribute to P_1_, while the sum of the CP of the DPV valve and the intracavitary pressure where the distal catheter is placed contribute to P_2_. For the sake of simplicity, we consider that the cavity in which CSF is shunted is the peritoneal cavity and therefore the intraperitoneal pressure (IPP) is the outlet pressure. An important concept to grasp is that in any DPV, the valve will be always closed when PP_shunt_ is ≤0, and it will open when PP_shunt_ is >0. Once the valve is opened, the CSF flow through the shunt will be regulated by Equation (2).

The definitions provided for opening pressure (OP) and CP in a DPV in many published papers are misleading. Most manufacturers report either the OP or the CP as required by the International Organization for Standardization (ISO), ISO 7197:2006, which regulates the minimum mechanical and technical requirements that manufacturers should report for implants used in the treatment of hydrocephalus [19]. OP is obtained from flow–pressure tests conducted in standardized testing rigs and is defined as the differential pressure above which flow of deaerated and deionized water through the valve starts at a fixed rate of 20 mL/h [19,20]. CP is the differential pressure below which the flow through the shunt ceases once the infusion pump in the testing rig stops [21]. In most valves, OP and CP are similar; therefore, the differences are clinically irrelevant. ISO 7197:2006 dictates that shunt manufacturers report the flow–pressure tests at 20 mL/h for their valves; they are classified as extra-low, low, medium, or high-pressure according to the pressure generated as this fixed flow rate. However, manufacturers usually report the flow–pressure data on a continuous scale from 5 to 50 mL/h (Figure 4).

## 6. Valve Mechanics and Technology

In any shunt assembly, the valve is the most important component because it regulates when the CSF flow starts and stops and it changes the amount of CSF fluid that passes through the shunt when intracranial pressure (ICP) is high. DPV can be classified according to the mechanical design in slit, miter, diaphragm, or spring-loaded “ball-in-cone” valves [22]. Most of the first generation and current DPV valves use the properties of the silicone elastomers and are of three different families: (1) slit, (2) miter, or (3) diaphragm valves [22]. Slit valves are silicone tubes closed at one end that have small cuts or slits in their wall (Figure 5A). When fluid pressure inside the catheter exceeds the predefined OP, which depends on the stiffness and the thickness of the tube wall, the slits open and CSF will flow [22]. The Chabbra^TM^ “slit n’ spring” valve (Surgiwear Limited, Shahjahanpur, India) is a low-cost shunt in which the slit valve is protected by a stainless steel spring [23] (Figure 6). A miter valve is basically a silicone tube that converges into opposed leaflets that opens when the pressure differential exceeds the OP of the valve, which, as in all silicone elastomers, depends on the size, shape, thickness, and length of the leaves [22]. The UltraVS™ Valve (Natus Medical Incorporated, USA) is an example of a miter valve. Diaphragm valves use flexible silicone membranes mounted in a plastic or metal seat upon which the diaphragm is moved by pressure differences (Figure 5B). When pressure differential is above the nominal valve OP, the diaphragm is displaced, and CSF flows around the diaphragm; when the pressure is below the OP, the diaphragm returns to its original position and closes the valve’s outlet [22]. The Contour-Flex™ valve (Natus Medical Incorporated, Middleton, WI, USA) is a typical diaphragm valve.

Spring-loaded ball-in-cone valves use a completely different approach to regulate pressure. These valves incorporate a metallic spring that exerts the manufacturer’s predefined force to a synthetic ruby ball located in a cone-shaped orifice [22]. When the differential pressure exceeds the OP threshold of the valve, the ruby ball is pushed against the spring and opens the outlet to allow flow (Figure 7). Conventional ball-in-cone DPV valves are the Codman-Hakim^TM^ Precision Fixed pressure valve (Integra LifeSciences Corporation, Plainsboro, NJ, USA) and the miniNAV^®^ (B. Braun Melsungen AG, Germany). Compared to silicone valves, ball-in-cone valves have a more precise OP and sustain higher CSF protein levels before becoming clogged than do silicone valves. Independent studies have shown that, in general, ball-in-cone valves have better mechanical characteristics than silicone valves and show a better convergence of pressure–flow performance when tested in vitro [20,24]. Despite the frequently seen notion that ball-in-cone valves “are less prone to the effects of the aging of materials than are miter or slit valves” [22], as far as we know, this has not yet been proved.
Figure 7(**Left**): Schematic drawing of a spring-loaded ball-in-cone mechanical valve. Arrow shows the direction of the flow. (**Right**): Figure 2 of the patent filed by S. Hakim [25] with a detailed drawing of the valve that the patent describes as “a pair of spring operated ball-type check valves made of metal or other relatively inert and non-toxic material not affected by temperatures in the surgical sterilization range. The twin valves are arranged in series to face in the same direction (outlet of one adjacent to the inlet of the other)” [25].
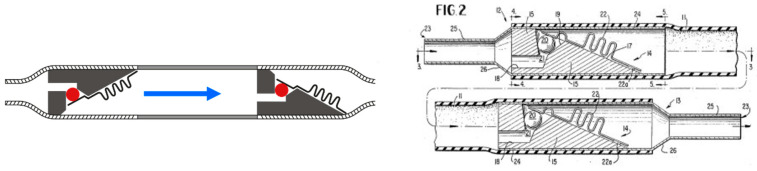


All of the silicone-based valves have the viscoelastic properties of the silicone elastomers; therefore, aging of these materials may significantly alter their performance. However, degradation and calcification of silicone catheters with deterioration of their mechanical properties, predisposition to catheter fractures, and migration of the different parts of the shunts have been reported only for catheters that are in direct contact with the subcutaneous tissue [26] but not in the silicon slits, miters, and diaphragms enclosed inside the housing of the valve.

## 7. Fixed Versus Adjustable Valves

Since the early 1990s, most manufacturers have produced both valves with fixed OP and adjustable (“programmable”) valves in which the OP settings can be modified noninvasively. The first available commercial adjustable valve (AV) was patented by Hakim in 1986 [27]; it is known today as the Codman–Hakim^®^ programmable valve (Integra LifeSciences Corporation). The Codman–Hakim^®^ is one of the best available adjustable valves, but it is also a good example of the limitations of such valves. In the earliest Hakim adjustable valve, the OP could be adjusted to 18 settings in small increments of 10 mmH_2_O by the application of an external magnetic field [27]. The most recent version of this valve allows adjusting the OP between 30 and 200 mmH_2_O (Integra LifeSciences Corporation).

There is still a fundamental misunderstanding of the reasons Hakim thought that an AV was required to manage hydrocephalus. AVs were not introduced for controlling shunt overdrainage, which is the wrong strategy for avoiding it, but to modify the OP to normalize the changes that an implanted shunt induces in ventricular size, IVP, and subdural stress [28,29,30]. In brief, Hakim biomechanically modeled the brain as an “open cell sponge” in which the venous blood and the extracellular fluid within the brain parenchyma can change their volume because both are connected to the extracranial venous system, which is in turn exposed to the atmosphere [29]. As stated by Hakim, when the ventricles increase their size in hydrocephalus, “the collapsibility of these compartments account for the bioplastic deformation of brain tissue” [29].

According to Hakim’s hypothesis, the part of the IVP that is not absorbed by the brain parenchyma is transmitted to the subdural space and is referred as subdural stress. Its magnitude is directly related to the ventricular size, and Hakim considered it one of the most important factors in hydrocephalus development [31]. In his words, once a shunt is implanted and “as soon as normal ventricle size has been reached, shunt revision should be made to raise CSF pressure in order to prevent further shrinkage of the ventricles”. The update of the OP and therefore of the IVP should restore subdural stress to normal [31]. Despite Hakim’s compelling theory, which pushed AV into the market, as far as we know, no clinical report has been published regarding this management technique.

Today, most neurosurgeons choose AV in iNPH patients for three main purposes: (1) controlling overdrainage by increasing the OP, (2) reducing the OP when improvement does not occur, and (3) partially closing the shunt to manage subdural effusions. However, as is discussed elsewhere, the use of AV for managing overdrainage is a mistaken strategy and, as Aschoff remarked, users that select it “can select different grades of over-drainage only, but not avoid it” [1]. In essence, any AV is a DPV with a variable OP. Therefore, these valves will open when PP_shunt_ is >0; however, when patients assume a standing or sitting position, the gravitational-induced HP will overcome the OP even at the highest OP settings (200 mmH_2_O = 14.7 mmHg). As described by Meier and Kintzel, they will “remain open too long when the patient moves into the upright position. In this way they could induce suction on the already atrophic brain” and therefore expose the iNPH patient to the adverse events induced by shunt overdrainage [32]. The reasons most patients with simple DPV tolerate chronic overdrainage are discussed in another paper of this Special Issue.

Some AV designs, such as the Codman–Hakim^®^ programmable valves, are sensitive to magnetic fields and therefore require routine reprogramming when an MRI is performed (Figure 8). Unintended changes in AV valve settings also occur when relatively low-intensity magnetic fields, such as those generated by tablet devices, cell phones, headphones, metal detectors, and others, are too close to the AV [33]. Many manufacturers have introduced “brakes” in their recent AV models to avoid unintentional reprogramming when the valves are exposed to magnetic fields [34]; however, these new valves generate important neuroimaging distortions (Figure 8), are expensive, and have not been tested in modern high-field magnetic field imaging (MRI) above 3 T.

However, the lack of available evidence proving the superiority of AV does not allow us to recommend adjustable over conventional DPV for managing NPH patients. This applies only to DPV alone and not to DPV with any additional “antisiphon” or “gravitational-control” devices. These “siphon-controlling” devices were introduced initially in 1973 by Portnoy et al. to reduce the incidence of subdural hematomas and effusions that are a consequence of shunt-induced subatmospheric intraventricular pressure due to the “shunt acting as a siphon, whose lower end vents to the atmosphere via the heart or peritoneal cavity at a significant distance below the foramen magnum” [35].

These “antisiphon” devices, independently of how they work—antisiphon devices or gravitational-compensating accessories—are discussed in another paper in this Special Issue. In our opinion, the combination of both DPV in series with devices to control shunt overdrainage when patients are standing should be the standard of care in most patients with hydrocephalus. However, to discuss the rationale for this recommendation is beyond the scope of this paper. In brief, in vitro studies have shown that antisiphon devices (ASD) reduce shunt overdrainage induced by the posture-increase in hydrostatic pressure when patients are standing or sitting [34,36]. In addition, some clinical studies have clearly shown that ASD reduce the number of adverse events in patients with iNPH and indirectly improve their clinical outcome [37,38,39].

Despite the lack of evidence, the guidelines for the treatment of iNPH published in 2005 concluded that “several retrospective studies suggest that adjustable (“programmable”) valves offer an advantage over fixed valves because corrections for under- or overdrainage, which are encountered relatively frequently with iNPH, can be performed noninvasively instead of requiring a surgical revision” [40]. However, all the studies reported in the 2005 guidelines were retrospective and do not support this recommendation when the Grading of Recommendations Assessment, Development and Evaluation (GRADE) working group criteria for evaluating the quality of evidence are applied [41]. Since the guidelines for the treatment of iNPH were published, some papers have evaluated the use of AV with or without antisiphon devices and have tried to compare patients with a gradual reduction of the valve OP vs. a fixed OP [38,42] or the optimal OP setting for these valves [43]. However, the design of these studies and/or their methodology precludes extracting solid conclusions from them.

In high-income countries, AVs double the cost of a fixed valve from the same manufacturer. However, in low- and medium-income countries, AVs have a significantly higher cost—in some countries, outrageously overpriced—than fixed valves, and therefore AVs are not affordable. However, despite the lack of robust evidence about their benefits, and when cost is not a relevant variable, it is important to take into consideration that AVs allow us to fine-tune the OP in many pediatric and adult patients with hydrocephalus and might reduce the number of shunt revisions.

## 8. The Resistance of Shunts and Valves: A Neglected Variable

Up to this point, we discussed how DPV valves open and close and the effect of the PP_shunt_ in regulating CSF flow. However, from Equation (2) it is obvious that when the valve in a shunt is opened, CSF flow is regulated by both PP_shunt_ and R_shunt_.

Interestingly, R_shunt_ is never reported by shunt manufacturers, and is available only from independent laboratories typically supported by research grants or academic institutions [44]. In Europe, the first independent laboratory that tested shunts was led by Aschoff in Heidelberg [45], and in the last 20 years, the most significant contributions have been made by the Cambridge Shunt Evaluation Laboratory, which conducted independent tests of all shunts available in the UK. The results were initially published by the Medical Devices Agency in the “blue reports” [20], but the Cambridge Laboratory has continued to publish independent evaluation for at least 26 commercially available shunts in peer-reviewed journals [44].

For a better understanding of the relevance of R_shunt_ in the management of hydrocephalus, in Figure 9 we plotted the maximum estimated flow of water through a constant 100 cm length silicone tube with a variable internal diameter (ID) of 0.40–1.20 mm at a DP from 0 of 60 cmH_2_O (0–44 mmHg). Most manufacturers provide distal catheters with an ID between 1.0 and 1.40 mm for VP/VA shunts, and some LP designs have catheters with an ID of 0.50–0.80 mm. As shown in this figure, following Poiseuille’s law, flow through a constant tube length decreases as internal diameter decreases.

For the sake of clarity, let us assume the estimated shunt flow in a real patient in whom all the variables that intervene in the regulation of DPV are known. This 72-year-old male had a nine-month history of gait disturbance, urinary incontinence, and nondetectable cognitive impairment. The patient’s height was 173 cm and his weight 82 kg (body mass index (BMI): 27.4, moderate overweight). ICP monitoring showed a mean ICP of 8 mmHg with ~40% high- and low-amplitude B-waves—wave amplitude > 10 or ≤ 10 mmHg, respectively—compatible with the diagnosis of iNPH [46]. If a neurosurgeon decided to implant in this patient a 100 cm length valveless silicone tube of 1.20 ID, the maximal possible CSF flow through the shunt when the patient is recumbent—assuming an IPP of 2 mmHg and a PP_shunt_ = 6 mmHg—would be ~209 mL/h. When this shunted patient stands (Figure 3), IVP is reduced to 0 mmHg or even to moderate negative values [47]. If we assume a gravitational-induced HP of 70 cmH_2_O (51 mmHg) when the patient is standing, the CSF flow could increase to a maximum of 1700 mL/h. In this simulated scenario, the patient exceeds the physiological production of CSF (~20 mL/h, 500 mL/24 h). In this situation, the cerebral ventricles will rapidly collapse, and the patient would be exposed to all the complications of shunt overdrainage. If another neurosurgeon had decided to implant a valveless silicone tube of the same length but with an ID of 0.60 mm (Figure 9), the same patient would have drained only 13 mL/h when recumbent, a value below the normal CSF production. However, when the patient is standing, the shunt will overdrain to a maximum CSF flow of 170 mL/h.
Figure 9Estimated in vitro flow through silicon tubes of 100 cm constant length and with an inner diameter (ID) varying from 0.4 to 1.2 mm at different differential pressures (DPs) (0–60 cmH_2_O; 0–44 mmHg). The reader can replicate these data by using the online calculator provided by vCalc™ (http://bit.ly/2Y9FEss) and considering that the viscosity of water at a temperature of 37 °C is 0.69 mPa-s [48]. A tube with an ID of 1.2 mm has a theoretical R_shunt_ of 1.72 mmHg/mL/min, of 2.43 mmHg/mL/min when the ID is decreased to 1.1, and of 139 mmHg/mL/min when the ID is 0.40 mm. At a DP of 60 cmH_2_O, the maximum flow for each tube will increase from 19 mL/h for a tube with an ID of 0.4 to 1539 mL/h for a tube with an ID of 1.2 mm. The reader needs to be aware that theoretical estimates in R_shunt_ reported here can differ from some published results on bench tests. Theoretical estimates may differ from results on tubes conducted in test rigs because of differences in the catheter length, presence of air bubbles, experimental temperature, composition of the infused fluid, and characteristics of the flow (pulsatile vs. steady).
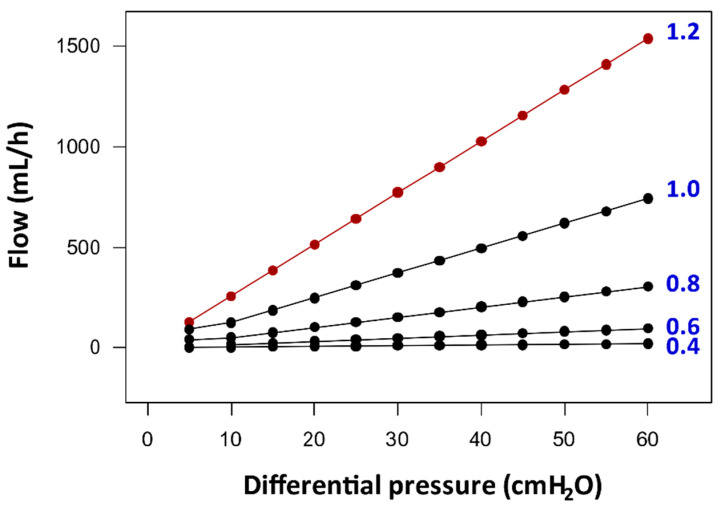


Based on this basic hydrodynamic principle, Sotelo, a Mexican neurosurgeon, introduced in 1993 a low-cost valveless shunt for managing patients with hydrocephalus [49,50]. In the most recent version, Sotelo’s shunt comprises a ventricular catheter, a connector, and a 100 cm length distal catheter with an ID of 0.51 mm opened at its end [49]. The rationale underlying this simple design is twofold: (1) to avoid the intermittent on–off working of DPV valves and to allow for a continuous CSF flow, and (2) to reduce hydrostatic-induced overdrainage when the patient is standing [51]. According to Sotelo, the CSF flow though this simple tube is close to the constant production of CSF (~0.35 mL/min, 21 mL/h) with modest variations in flow imposed by the gravity-induced HP generated by postural changes [51]. Sotelo’s shunt can be considered as a simple “flow-regulated” shunt. Valveless shunts should be used only in special circumstances in which they can play an important role, especially in treating isolated IV ventricles, multicompartmental hydrocephalus, isolated CSF collections, or in treating arachnoidal cysts. However, neurosurgeons using them need to know the relevance of the ID to avoid both shunt under- and overdrainage.

The previous examples reflect the need to consider the resistance of all the components of a shunt assembly and, ideally, to have, at least, good estimates of the IVP and IPP when managing individual patients with iNPH. R_shunt_ and R_valve_ are not provided by most shunt suppliers and they need to be estimated from the flow–pressure data provided (Figure 4). However, this neglected variable defines the shunt flow when PP shunt is ≥0 mmHg, and therefore is an essential information that neurosurgeon should have. This was clearly shown in the seminal but undercited paper published by Rayport and Reiss in 1969 [14], in which they reported a 17-month-old girl with a communicating hydrocephalus deteriorating to a coma and who recovered after the insertion of an external ventricular drainage. The CSF outflow in this girl exceeded 1000 mL/24 h at an OP of 175 mmH_2_O (12.8 mmHg) [14]. In this patient, a Pudenz–Heyer ventriculo-atrial shunt was implanted with a CP of 70 mmH_2_O (5.1 mmHg), but the external ventriculostomy was maintained for controlling the CSF output. The patient deteriorated again after surgery, and the CSF outflow at neuroworsening was above 25 mL/h despite a functional shunt. To manage this problem, the girl required a second Pudenz–Heyer shunt implanted in parallel to drain the excessive CSF production that a single shunt was unable to manage [14].

In a second part of their study directed toward understanding the problem they faced treating this unusual case, Rayport and Reiss designed an in vitro setup in which they tested the hydrodynamic profile of three different valves (Spitz–Holter, Pudenz–Heyer, and Cordis–Hakim) with the same reported nominal CP range [14]. The pressure–flow results shown in their Figure 4 (page 464 of the manuscript) showed significant differences in the hydrodynamic behavior and significant difference in the maximum flow rate achievable by the three DPVs. In brief, Rayport and Reiss showed that the R_shunt_ of the three similar medium-pressure DPVs were significantly different: 2.4 mmHg/mL/min for the Cordis–Hakim, 4.2 for the Pudenz–Heyer, and 16.6 mmHg/mL/min for the Spitz–Holter (values estimated by us from the Figure 4 of the manuscript) [14].

## 9. Hydrodynamic Properties of Different Shunts

The relevance of the R_shunt_ in the literature, raised indirectly by Rayport and Heiss’s seminal paper, was largely neglected in the neurosurgical literature and reassessed only after the studies of independent bodies such as the Aschoff and Cambridge shunt laboratories were published [24,52]. Czosnyka et al., in their original “blue cover reports” and in subsequent studies, defined valve hydrodynamic resistance as the regression line obtained in pressure–flow tests [24]. In their comprehensive reports, they have shown that most DPVs have an R_shunt_ much lower (<2–5 mm Hg/mL/min) than the normal physiological values quantified by infusion tests, which are in the range 6–10 mmHg/mL/min [24].

In Table 1, we summarize the R_shunt_ values of some DPVs with and without the distal catheter attached, using the results reported by the Cambridge Shunt Laboratory in various publications [20,24,53,54,55,56,57]. The single most important caveat for neurosurgeons managing patients with iNPH is that most shunt devices available have a low R_shunt_ even with the attached distal catheter, and therefore overdrainage will be induced depending on the variables already discussed. A second message is that different valves included by manufacturers in the same category (i.e., medium-pressure DPVs) have variable OP and significantly different hydrodynamic profiles that should be known when managing difficult cases or reconsidering shunt revision in nonresponsive iNPH patients in which shunt underdrainage is suspected.
jcm-10-01210-t001_Table 1Table 1The information shown in this table regarding resistance was extracted from different papers and reports published by the UK Shunt Evaluation Laboratory and not from the information provided by manufacturers [53]. When not available from independent studies, CP reflects the manufacturer information. Sotelo’s shunt data are from Sotelo et al. [51,58]. The hydrodymanic profiles of silicone tubes are calculated with the online calculator provided by vCalc™ (http://bit.ly/2Y9FEss, accessed on 1 September 2020) and considering that the viscosity of water at a temperature of 37 °C is 0.69 mPa-s. CP: closing pressure in mmHg. R_valve_: static resistance of the valve calculated from pressure–volume tests and measured in mmHg/mL/min. R_shunt_: total resistance of the shunt as provided by the manufacturer that includes the **R_valve_** and the distal catheter provided by the manufacturer attached. Delta valve (Medtronic Medical Devices, Minneapolis, MN, USA) have two performance levels (PL1 and PL2) [59]. The values refer to the R_valve_ without the siphon-controlled device active [53]; Codman–Hakim^TM^ (Integra Lifesciences, Plainsboro, NJ, USA) [20]; Polaris (SOPHYSA, Orsay, France).ModelCP(mmHg)R_valve_(mmHg/mL/min)R_shunt_(mmHg/mL/min)Silicone 100 cm (1.2 mm ID)0No valve1.72Silicone 100 cm (1.0 mm ID)0No valve3.57Silicone 100 cm (0.6 mm ID)0No valve27.5Sotelo’s shunt (0.51 mm ID)0No valve52.7Codman–Hakim (low)2.91.654.92Codman–Hakim (medium-low)5.11.835.10Codman–Hakim (high)9.62.205.46Delta valve (PL1)1.91.93.6Delta valve (PL2)4.02.23.7Contour Flex (low)0.7–3.72.33.5Contour Flex (medium)3.7–8.12.53.8Contour Flex (high)8.1–12.5 2.84.2Polaris (30 mmH_2_O)2.21.64.6Polaris (70 mmH_2_O)5.142.15.1Polaris (110 mmH_2_O)8.12.55.6Dual Switch Valve (5 cmH_2_O)3.72.22.9Dual Switch Valve (10 cmH_2_O)7.42.22.9

## 10. DPV and Shunt Assemblies for Complex Hydrocephalus Patients

Patients with hydrocephalus are a heterogeneous population. In their management, there is a subgroup of complex cases in which an adequate shunt selection and in-series or integrated devices (i.e., antisiphon or gravitational-control devices) are much more relevant than in conventional patients to reduce the risk of overdrainage and shunt malfunction, and to improve their outcome. Some cases of pediatric hydrocephalus or iNPH, post-traumatic hydrocephalus and the entity defined by Oi in 1996 named long-standing overt ventriculomegaly in adults (LOVA) [60] are some of these cases [61,62,63]. In brief, LOVA is a type of hydrocephalus that develops during childhood, but where symptoms appear during adulthood and in most patients is associated with aqueductal stenosis, massive ventriculomegaly, and macrocephaly [61]. Many have a clinical presentation compatible with NPH syndrome [61].

Most of these complex patients share a significant increase in ventricular size (I. Evans > 0.42) and a significant disproportion between the intracranial and the cerebral volume—with or without macrocephaly. In these patients, ICP monitoring has shown different profiles, but, frequently, these patients have a mean ICP < 12 mmHg [62]. Oi et al. reported that patients with LOVA have a high ICP [61]. However, in the paper by Oi et al., ICP statistics were not presented and their example of a “high ICP” presented in Figure 4 shows a mean ICP of ~12 mmHg, with a train of high amplitude B-waves with peaks around 30 mmHg [61]. This profile is compatible with the most frequent pattern found in iNPH patients and described by Sahuquillo et al. as “compensated hydrocephalus” [46], and also found as the predominant ICP profiles in patients with severe ventriculomegaly and macrocephaly [62]. There is the generalized idea that in these LOVA or LOVA-like patients—severe ventriculomegaly not associated with aqueductal stenosis—the risk of implanting a conventional DPV or even AV is very high. This is based on the daily practice of many neurosurgeons, supported by reports in the literature, where the incidence of subdural hematomas/effusions in these patients is very high [61]. Due to the unusually high risk of complications Oi et al. suggested selecting neuroendoscopic surgery as the initial treatment for LOVA patients [61].

However, in treating LOVA or LOVA-like patients, we support the idea of Kiefer et al. that for these patients the best treatment option is the combination of an AV with a gravitational device [64]. In their seminal paper, Kiefer et al. stated that “This study demonstrates that the gravitational shunts—as opposed to conventional DP shunts—can effectively prevent over-drainage even in this high-risk group with LOVA hydrocephalus” [64]. Selecting a high OP when treating these patients increases the risk of underdrainage when the patient is recumbent and does not reduce the risk of overdrainage when the patient in sitting or standing. The design and rationale of the different antisiphon devices and their indications are discussed in another paper in this Special Issue.

## 11. Conclusions and Recommendations

It is obvious that any DPV is unable to control for shunt overdrainage when implanted without any gravitational-control or antisiphon device in patients without a significant increase in the IPP. This topic is discussed in another part of this Special Issue. However, devices to control for postural-induced changes in hydrostatic pressure are not yet generally used in adult and pediatric neurosurgery.

To close this review, we would like to give some recommendations for selecting the most appropriate shunt for a patient with iNPH. The reader needs to take into consideration that our recommendations are not evidence-based because of the lack of controlled trials and that they are exclusively based on the quantitative assessment of CSF dynamics and the reports elaborated from independent bodies regarding valves and shunts. We have also incorporated the experience of the senior authors in monitoring many patients with iNPH before and after shunting [65,66]. Although we use these recommendations routinely in the management of iNPH, we cannot demonstrate the relevance of them in modifying the outcome of iNPH patients until tested in adequately designed prospective randomized–controlled trials. However, we believe that these recommendations could be helpful for neurosurgeons to rationalize the management of complex hydrocephalus cases and in avoiding the application of one-size fits all simple solution for a complex problem.

In patients with iNPH and unknown IVP, because ICP is not available or used, it is recommended to use low-pressure valves. The use of these valves will avoid problems with underdrainage when the patient is recumbent. Selecting high-pressure DPVs does not control overdrainage and increases the risk of underdrainage while the patient is recumbent.Spring-and-ball mechanical valves are much more reliable in OP than are silicone-based valves. In Table 1, we show that the OP can be defined with a narrow limit in mechanical valves, but that manufacturers usually provide ranges of “working pressures” for nonmechanical valves, reflecting the potential variability in OP even in the same lot of valves.Without gravitational control of the postural-induced hydrostatic pressure, all DPVs, independent of their OP, will overdrain when the patients assume the standing or sitting position. An exception to this rule is when for different reasons—obesity, pregnancy, abdominal distension, etc.—IPP is significantly raised, causing PP_shunt_ to be reduced when the patient is standing. Therefore, neurosurgeons should consider gravitational control—either integrated in the valve or as an added device—in most patients in which DPVs are used. As Czosnyka et al. cautioned, neurosurgeons should take care when implanting “low-resistance shunts” into patients with gross ventricular dilatation or greatly increased cerebrospinal compliance, and of their subsequent mobilization, if no siphon control mechanism is added [24].In the selection of DPVs, we recommend knowing the specifications provided by the manufacturer and the hydrodynamic properties of the selected shunt from independent research. It is important to remember that a high resistance valve together with a catheter of reduced ID increases the risk of nonresponse in iNPH patients.The most rational approach to selecting the most appropriate shunt for an individual patient is to have the best information concerning shunt hardware and as many patient variables as possible that will influence PP_shunt_. Apart from height, weight, and BMI, an estimation of the mean ICP and IPP is helpful [67].In low- and medium-income countries, AVs are significantly overpriced and therefore not affordable. However, despite the lack of robust evidence about their clinical benefits, and when cost is not a relevant variable, AVs allow us to fine-tune the OP in many pediatric and adult patients with hydrocephalus and might reduce the number of shunt revisions.An important recommendation for all neurosurgeons involved in the management of iNPH was introduced in the *Shunt Book* published by Drake and Sainte-Rose: “Even if the ideal shunt existed, it would be quickly rendered useless by improper surgical technique during implantation” [6].

## Figures and Tables

**Figure 1 jcm-10-01210-f001:**
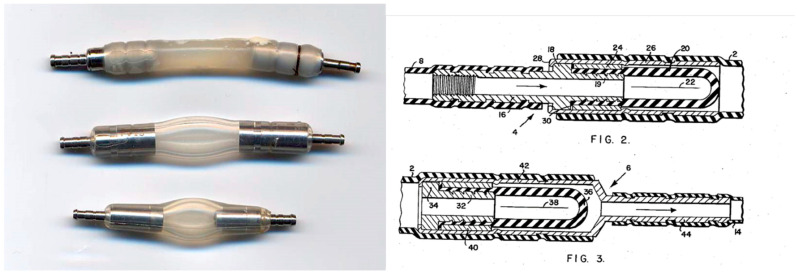
(**Left**). Reproduced with permission from A. Aschoff. Three different Holter valve designs are shown. The first valve on top is the oldest model (circa 1970), more recent designs in the middle and on the bottom. This valve was discontinued and no longer available. (**Right**). Reproduced from Figure 2 and Figure 3 of the patent filed 2 October 1956 by Holter and Spitz [12]. Both figures are enlarged sections showing the inlet (FIG. 2) and outlet valves (FIG. 3) of the device. Labels 22 and 38 show the inlet and outlet tubes with lateral slits “three-sixteenth inch in length” (0.80 cm), which functions as the actual valve.

**Figure 2 jcm-10-01210-f002:**
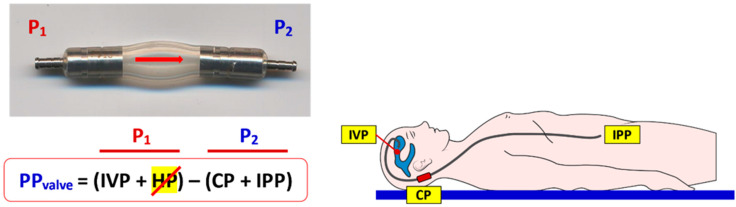
(**Top**). In this figure, the pressures defining the perfusion pressure of the valve (PP_valve_) are shown. P_1_ is the pressure at the inlet of the valve and P_2_ at the outlet. P_1_ is the sum of the intraventricular pressure (IVP) and the gravitational-induced hydrostatic pressure (HP). P_2_ is the sum of the closing pressure (CP) of the valve and the intraperitonal pressure (IPP) if the distal catheter is placed in the peritoneum or the intra-atrial pressure (IAP) if the distal tip is placed within the right atrium. (**Bottom**). When the shunted patient is recumbent, the hydrostatic difference between the tips of the ventricular and intraperitoneal catheters is negligible and ~0 mmHg; therefore, the HP in the equation that calculates the PP_valve_ can be removed.

**Figure 3 jcm-10-01210-f003:**
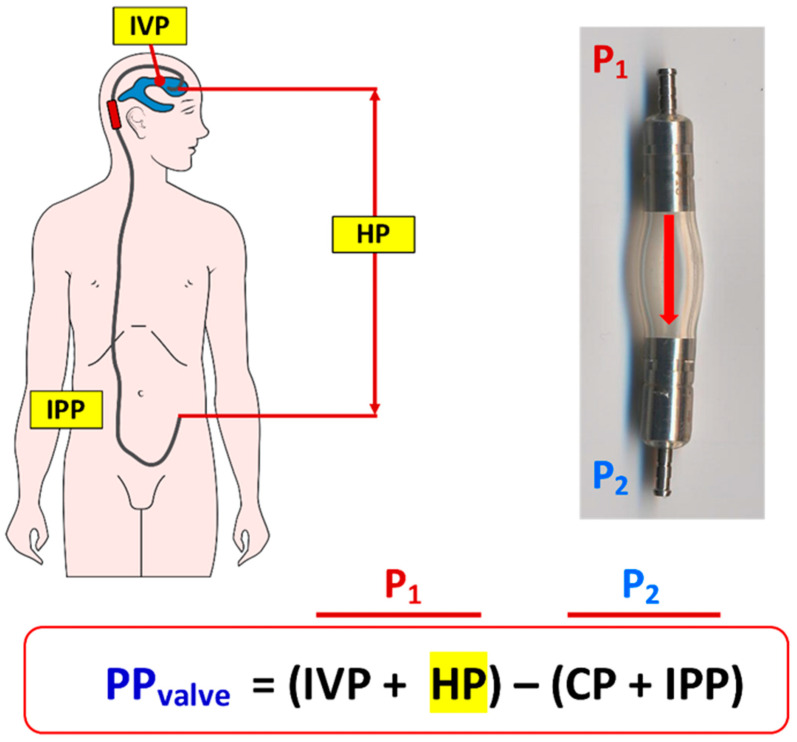
This figure shows the changes in PP_valve_ when a shunted patient is standing. In this scenario, the IVP is ≤0 mmHg and the gravitational-induced HP is equal to the height between the tips of the ventricular and distal catheters. In a 173 cm height adult, this difference can be equal to 70 cm, which will generate a gravitational HP of 51.2 mmHg. If the patient has an implanted differential-pressure valve (DPV) with an opening pressure (OP) of 5 cmH_2_O (3.7 mmHg) without any gravitational or antisiphon device, and assuming an IPP of 0–3 mmHg, these conditions will generate a PP_valve_ in a standing patient of ~44 mmHg and therefore the valve will be permanently open when patient is sitting or standing.

**Figure 4 jcm-10-01210-f004:**
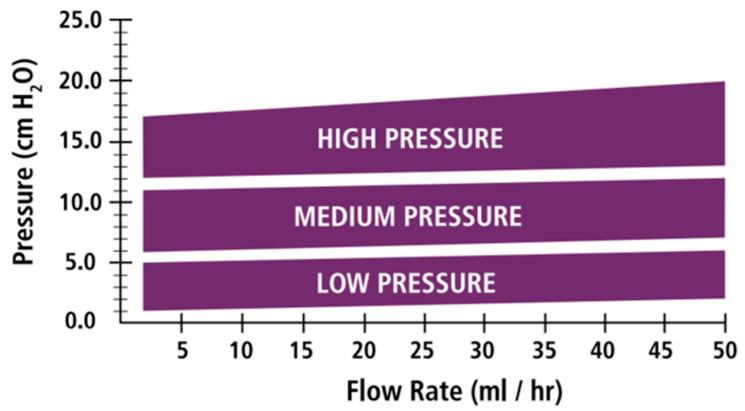
Figure reproduced with permission from the Natus Neurosurgery and Neurocritical CareProduct and Accessory Catalog, 2019 (https://natus.com, accessed on 23 June 2020). In this graph, the flow–pressure data for the Contour-Flex™ family of valves are shown. Contour-Flex™ are typical silicone diaphragm valves with a low–medium- and high-pressure ranges. Flow rates in the x-axis represent a variable flow rate from 1 to 50 mL/h while in the y-axis the pressure generated at each flow rate is shown. Colored bands define the variability of the pressures obtained for each flow rate.

**Figure 5 jcm-10-01210-f005:**
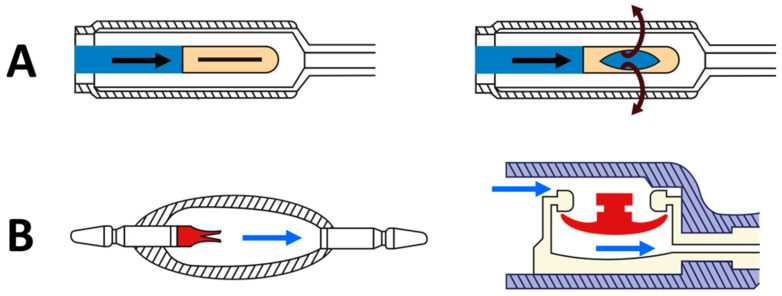
(**A**). Schematic drawing of a silicone slit valve. Slit valves are essentially silicon tubes that are closed at the end and have slits in their sides. On the left, a closed tube with a lateral slit is shown. On the right, the slit opens and allows CSF flow when the pressure inside the tube exceeds the predefined OP setting defined by the manufacturer. Arrows indicate the direction of CSF flow. (**B**). On the left, an opener miter valve is shown. The two silicon leaflets open when CSF inside the shunt exceeds the predefined OP. (**B**). On the right, a drawing of a diaphragm valve is shown. When CSF in the circuit exceeds the OP of the valve, the diaphragm moves down and allows the fluid to pass through. Arrows indicate the direction of the flow.

**Figure 6 jcm-10-01210-f006:**
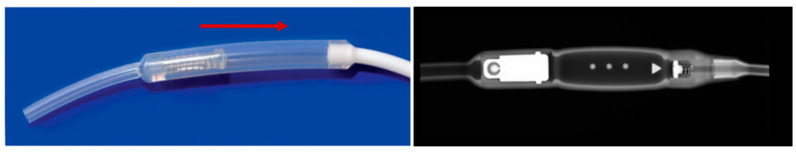
(**Left**). The Chabbra^TM^ “slit n’ spring” valve (Surgiwear Limited, Shahjahanpur, India) is a design based in the slit valve. In this case, the silicone tube within the flushing reservoir in which the slit seat is protected by a stainless-steel spring. The red arrow indicates the direction of the flow. (**Right**). Radiograph of a Codman–Hakim^TM^ Precision low-medium DPV (Integra LifeSciences Corporation). This DPV is a conventional mechanical spring-loaded ball-in-cone valve. The three dots indicate the OP of the valve and the short arrowhead the direction of the flow.

**Figure 8 jcm-10-01210-f008:**
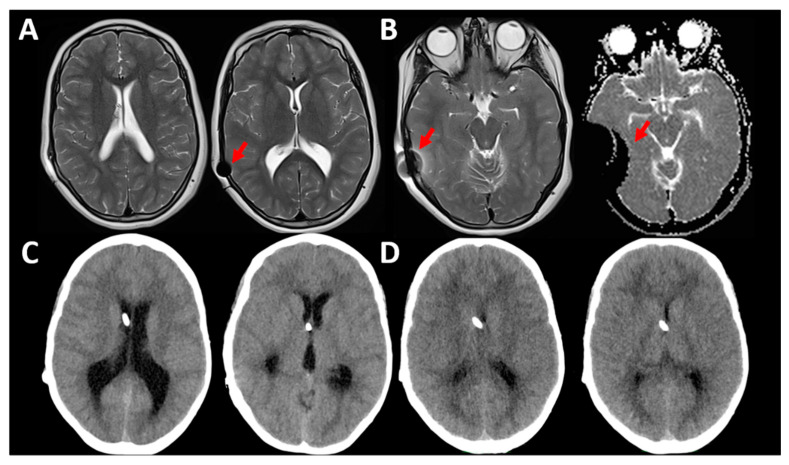
This 13-year-old girl had surgery for a Chiari 1 malformation with hydrocephalus. She had a Hakim programmable valve implanted with an OP of 70 mmH_2_O. The patient was scanned in a 1.5 T MRI for control (**A**,**B**). In (**A**), the ventricular size was normal. Arrows show the magnetic artifact produced in the different MR1 sequences that was moderate in T2-weighted images and very strong in the diffusion-weighted sequences (**B**). The patient’s valve was accidentally reprogrammed by the doctor on call to an OP of 150 mmH_2_O immediately after the MRI. She was admitted three days later because of headache and drowsiness. The CT scan on admission (**C**) showed a significant increase in the ventricular size. The OP of the valve was readjusted to 70 mmH_2_O. The patient improved and the new CT scan at discharge is shown in (**D**).

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
