# Peer review of "How to Choose a Shunt for Patients with Normal Pressure Hydrocephalus: A Short Guide to Selecting the Best Shunt Assembly"

_jcm, 2021, doi:10.3390/jcm10061210_

Round 1

Reviewer 1 Report

I feel that overall this paper nicely and succinctly summarizes the history of neurosurgical management of hydrocephalus (particularly as it pertains to the development of shunts and valve types). In addition, it focuses on a comprehensive review of CSF dynamics and various DPV and adjustable valve types.  It accomplishes the above very nicely and is well-written and a wonderful addition to the literature. 

I do feel that the conclusions drawn with regards to NPH are a bit weak. The authors acknowledge these limitations and the lack of studies with regard to NPH. However, it would be helpful to include a review of the literature citing the management of NPH comparing different valve types. Alternatively, I might recommend removing the part about NPH and solely focusing on the review of valve types and CSF dynamics. 

Reviewer 2 Report

The authors provide a challenging up-to-date review on the theoretical basis of intraventricular shunts with particular emphasis on their use in normal pressure hydrocephalus. They introduce the mathematical descriptions of the determinants of flow of the cerebrospinal fluid via the shunt to the peritoneal cavity and provide experimental data for optimal properties of the catheter. The paper is nicely illustrated. It is of general importance for neurosurgeons and beyond to everyone who is interested to understand the benefits and hazards of such devices for draining a communicating hydrocephalus. Above all, the review is of relevance for patients suffering from normal pressure hydrocephalus as the pathophysiology of this disorder is still a topic of investigation.

A few issues ought to be improved.

  1. In the first sentence of the abstract the authors mention endoscopic third ventriculostomy (ETV). But during the article they do not provide any further details about it. Specifically, the equations they present do not apply to ETV. ETV has undoubtedly a place in aqueduct stenosis or occlusion and in communicating hydrocephalus. But its place in normal pressure hydrocephalus is ill-defined and, in fact, unsettled. Therefore, the authors should delete it from the paper.
  2. The authors refer to the seminal work by Rayport and Reiss several times throughout the paper. Unfortunately, they refer to reference 15 concerning this publication (p 4) instead of 17. Also, they cite the second author systematically as Heiss instead of Reiss. Please, correct.
  3. There are problems with misses in the reference list (e.g. 28, 29).

Reviewer 3 Report

This is a well-written valuable review guiding shunt selection in iNPH.

The only issue I surely criticize is the attitude toward adjustable valves (page 11, line 364-374). I argue that favoring fixed-pressure valves in iNPH is a step-back. Although the evidence favoring adjustable valves is only retrospective, I see a slavery requirement of RCT between adjustable and fixed pressure valves (to indicate the primality of either of them) a bit rudimentary. There are fixed pressure and adjustable valves that are otherwise similar in flow features (e.g. PS Medical Delta and Strata) indicating that it’s easy to show decrease in need of valve revisions when using adjustable valves (e.g. in our clinic at least one out of three revisions in iNPH patients were avoided). Furthermore, in the welfare states, the price difference (600 vs 900 €) is practically negligible. However, the overall cost-effectiveness (including need for adjustments after MRI-imaging in non-MRI safe adjustable valves), should be calculated. This is possible to do without RCT when using correct shunt configuration in prospective controlled study or comprehensive quality register.

Any comment on the ventricle size and optimal shunt? E.g. still, at least in some clinics, LOVA patients may be treated as iNPH patients. Traditionally, when shunting patients with very large ventricles, a high opening pressure is suggested.

In addition, some caveat on the applying the measurable physiological parameters and equations could be discussed. Sometimes in real life, we can see biological phenomena that are hard to explain by the fluid physics only.

Reviewer 4 Report

Well structured  and well organized work.The Authors have dealt with appropriateness and competence a subject so complex that historically it has been one of the most  discussed points in the neurosurgical literature.I believe that the conclusions are appropriate
